# Evaluating CDK4/6 Inhibitor Therapy in Elderly Patients with Metastatic Hormone Receptor-Positive, HER2-Negative Breast Cancer: A Retrospective Real-World Multicenter Study

**DOI:** 10.3390/cancers16203442

**Published:** 2024-10-10

**Authors:** Palma Fedele, Matteo Landriscina, Lucia Moraca, Antonio Cusmai, Antonio Gnoni, Antonella Licchetta, Chiara Guarini, Laura Lanotte, Maria Nicla Pappagallo, Assunta Melaccio, Guido Giordano, Felicia Maria Maselli, Antonello Pinto, Francesco Giuliani, Vincenzo Chiuri, Francesco Giotta, Gennaro Gadaleta-Caldarola

**Affiliations:** 1Oncology Unit, “Dario Camberlingo” Hospital, 72021 Francavilla Fontana, Italy; chiara.guarini@asl.brindisi.it (C.G.); antonello.pinto@studenti.unimi.it (A.P.); 2U.O. Medical Oncology and Biomolecular Therapy, Department of Medical and Surgical Sciences, University of Foggia, 71100 Foggia, Italy; guido.giordano@unifg.it (G.G.); felicia.maselli@unifg.it (F.M.M.); 3Oncology Unit, “Teresa Masselli Mascia” Hospital, 71100 San Severo, Italy; lucia.moraca@aslfg.it; 4Oncology Unit, I.R.C.C.S. “Giovanni Paolo II”, 70124 Bari, Italy; a.cusmai@oncologico.bari.it (A.C.); f.giotta@oncologico.bari.it (F.G.); 5Oncology Unit, “Sacro Cuore di Gesù” Hospital, 73014 Gallipoli, Italy; antonio.gnoni@aslle.it (A.G.); antonella.licchetta@aslle.it (A.L.); vincenzo.chiuri@aslle.it (V.C.); 6Oncology Unit, “Mons. A. R. Dimiccoli” Hospital, 70051 Barletta, Italy; laura.lanotte@aslbat.it (L.L.); gennaro.gadaleta@aslbat.it (G.G.-C.); 7Oncology Unit, “San Paolo” Hospital, 70123 Bari, Italy; marianicla.pappagallo@asl.bari.it (M.N.P.); assunta.melaccio@asl.bari.it (A.M.); francesco.giuliani@asl.bari.it (F.G.)

**Keywords:** CDK4/6 inhibitors, elderly patients, metastatic breast cancer, hormone receptor-positive, HER2-negative, real-world data, retrospective study

## Abstract

**Simple Summary:**

This study explores the effectiveness and safety of a common breast cancer treatment in elderly patients, a group often overlooked in clinical trials. Specifically, it focuses on using CDK4/6 inhibitors with hormone therapy for treating metastatic HR+/HER2- breast cancer in patients aged 70 and older. This research aims to understand how well this treatment works and how tolerable it is for this vulnerable population, which faces unique challenges due to age and other health conditions. The results of this study may provide valuable insights for clinicians, enabling more informed treatment decisions which optimize therapeutic outcomes and enhance the quality of life for elderly patients with breast cancer.

**Abstract:**

Background: Metastatic HR+/HER2- breast cancer is commonly treated with CDK4/6 inhibitors in combination with endocrine therapy. However, the efficacy and safety of this approach in elderly patients (≥70 years) remain unclear, particularly in the context of real-world clinical practice. This study aims to evaluate the clinical outcomes and tolerability of CDK4/6 inhibitor treatments in this fragile population, which is often under-represented in randomized clinical trials. Patients and methods: This retrospective multicenter study included elderly patients with metastatic HR+/HER2-negative breast cancer receiving first-line CDK4/6 inhibitors. The primary endpoint was progression-free survival (PFS). The secondary endpoints focused on the overall survival (OS), safety, and tolerability, considering variables such as tumor subtype, age, comorbidities, and treatment specifics. Results: The median PFS and OS were slightly lower than those reported in clinical trials, reflecting the inclusion of a more fragile population. The luminal B subtype was linked to a poorer PFS, while other factors like age, BMI, and ECOG status did not significantly affect the outcomes. A safety analysis indicated a higher incidence of grade 3 or higher toxicities, especially in frail patients, leading to dose reductions. Despite these challenges, CDK4/6 inhibitors were generally well-tolerated, allowing most patients to continue therapy. Conclusions: CDK4/6 inhibitors with endocrine therapy are effective in elderly patients with metastatic HR+/HER2- breast cancer, though careful management is crucial to balance efficacy and minimize adverse events.

## 1. Introduction

CDK4/6 inhibitors (abemaciclib, palbociclib, and ribociclib) combined with endocrine therapy (ET), which includes aromatase inhibitors and fulvestrant, with or without LHRH agonists, are the standard of care for the first-line treatment of patients with HR+/HER2-negative metastatic breast cancer, except for those experiencing a visceral crisis [1,2,3,4].

Clinical trials, in which progression-free survival (PFS) is consistently the primary endpoint, have demonstrated a significant benefit for CDK4/6 inhibitors compared to ET alone in both premenopausal and postmenopausal women. While the improvement in the PFS was significant across all studies, the overall survival (OS) showed significant benefits only in a subset of these trials [5,6,7,8,9,10,11,12].

The treatment of hormone receptor-positive (HR+) metastatic breast cancer (MBC) has significantly improved due to a better understanding of endocrine resistance mechanisms and the development of targeted therapies. New oral SERDs, along with inhibitors such as mTOR, PI3K/AKT, and PARP, have expanded the treatment options and improved survival outcomes for this patient population. Additionally, novel drugs like antibody–drug conjugates (ADCs) and PROTACs are showing clinical efficacy, even in cases of endocrine resistance. These advances bring new challenges, such as optimizing treatment sequencing, integrating ADCs into precision medicine, and using predictive biomarkers to guide therapy choices [13].

The elderly population presents unique challenges in cancer management due to clinical complexity and associated comorbidities. Treatment decisions are often based on chronological age rather than functional condition, leading to overtreatment in less fit patients and undertreatment in those who could benefit from comprehensive oncological management. Although one-third of breast cancer cases are diagnosed in patients over the age of 70, these patients are frequently under-represented in clinical trials, and those who participate tend to be fitter than the general elderly population. This creates a gap in clinical knowledge and limits the ability to optimize therapies for this specific group [14].

Concerns regarding the use of CDK4/6 inhibitors in elderly patients include the potential risk of adverse events and issues with treatment adherence and compliance. However, studies such as PALOMA 2, PALOMA 3, MONALEESA-3, MONALEESA-2, MONARCH-3, and MONARCH-2 have demonstrated significant benefits in terms of PFS and OS, suggesting that these therapies may also be effective for elderly patients, although further research is needed.

There are currently limited data, mostly from exploratory analyses, on the efficacy of new combinations involving CDK4/6 inhibitors, such as those with tamoxifen, oral SERDs, or PIK3CA/AKT/mTOR inhibitors, especially in elderly patients. These agents are still under investigation in clinical trials [15].

To address the need for specific data and optimize therapies in elderly patients (age ≥ 70 years) with HR+/HER2- metastatic breast cancer, we conducted a multicenter observational retrospective analysis. This study focuses on CDK4/6 inhibitor therapy in older women, aiming to generate data which can improve clinical management and fill the current knowledge gap. This study aimed to assess the effectiveness and safety of first-line CDK4/6 inhibitors in women aged 70 and older with metastatic breast cancer. The primary objective was to evaluate the PFS, with secondary outcomes including the overall survival (OS) and safety, assessed by monitoring adverse events throughout the treatment period.

## 2. Materials and Methods

### 2.1. Study Design and Patients

We gathered data from elderly patients diagnosed with metastatic hormone receptor-positive (HR+)/HER2-negative (HER2-) breast cancer, who began CDK4/6 inhibitor therapy between 1 January 2020 and 15 May 2024, across multiple centers. Eligible patients were 70 years or older and received CDK4/6 inhibitors as their initial treatment for metastatic disease, though prior treatment for localized breast cancer was permitted.

The collected information included sociodemographic factors (age, comorbidities), clinical characteristics (performance status, body mass index), pathological data (histology), intrinsic breast cancer subtypes or molecular classification [16,17], details of metastases (location), type of hormone therapy (fulvestrant, aromatase inhibitors), and the type of CDK 4/6 inhibitor. The geriatric assessment was performed using the G8 score. All patients presented with HER2-negative disease, as HER2 protein overexpression or gene amplification was an exclusion criterion for the study. However, since this analysis was conducted prior to the identification of HER2-low as a distinct category, HER2 status was recorded as either positive or negative in the patient records. Therefore, information regarding HER2 2+, 1+, or 0 expression levels is not available for this cohort.

CDK4/6 inhibitors were administered as per the standard dosing regimens for each specific drug, continuing until disease progression, intolerable toxicity, or death. None of the patients had received CDK4/6 inhibitors in the adjuvant setting.

In adherence to the European General Data Protection Regulation (GDPR) (2016/679), participants’ privacy was safeguarded by assigning each subject a unique anonymous numerical code.

This unique patient code enabled electronic linkage across different databases. All the results were presented in aggregated form, ensuring that data were not attributable to any single institution, department, physician, or individual prescribing behavior. This study underwent review by the Ethics Committee of the Puglia region.

### 2.2. Study Endpoints

The efficacy endpoints of this study included the following: progression-free survival (PFS), defined as the duration from the first administration of a CDK 4/6 inhibitor to the first documented progression or death from any cause, whichever occurred first; and overall survival (OS), defined as the time from the first administration of a CDK 4/6 inhibitor to death from any cause. Patients without progression at the last follow-up were considered censored for PFS, while patients alive at the last follow-up were considered censored for both OS and PFS. Safety was assessed by recording adverse events (AEs) related to CDK 4/6 inhibitors, evaluated using the National Cancer Institute Common Terminology Criteria for Adverse Events (NCI-CTCAEs), version 5.0.

### 2.3. Statistical Analysis

The characteristics of the sample were described using descriptive statistics, including medians with ranges for the continuous variables and absolute frequencies and percentages for the categorical variables. The PFS and the OS, along with the median follow-up periods for these endpoints, were evaluated using reverse Kaplan–Meier estimates, with the results presented as median values and 95% confidence intervals (CIs). Multivariate analysis was performed to assess the impact of various clinical factors on PFS. In this analysis, PFS was considered the primary outcome, with several prognostic factors evaluated as explanatory variables. These factors included age, body mass index (BMI), luminal subtype, presence of visceral disease, G8 score, Eastern Cooperative Oncology Group (ECOG) performance status, type of hormone therapy, and type of CDK 4/6 inhibitor. Statistical significance was defined as a *p*-value < 0.05.

AEs related to treatment were classified according to preferred terms (PTs) and organized by the primary System Organ Class (SOC) using the Medical Dictionary for Regulatory Activities (MedDRA) thesaurus, version 23.0. For AEs occurring more than once in the same patient, the highest grade according to the NCI-CTCAE criteria was reported. The analysis included all patients who had started treatment with a CDK 4/6 inhibitor. Statistical analyses were conducted using the R statistical software platform (version 4.4.1).

Given the retrospective nature of this study, a formal power analysis was not conducted. The sample size was determined by the number of eligible patients available across the participating centers, reflecting real-world clinical practice.

## 3. Results

The demographic and clinical characteristics are presented in Table 1. The cohort consisted of 160 patients who initiated CDK4/6 inhibitor therapy across seven Italian sites between January 2020 and May 2024. The patients’ ages at breast cancer diagnosis ranged from 70 to 85 years, with 51.26% diagnosed at or before 75 years of age, and 48.74% diagnosed after 75 years. Among the participants, 68.13% had de novo stage IV disease at diagnosis, while 31.87% were initially diagnosed at stages I to III.

At the start of CDK4/6 inhibitor therapy, all patients presented with metastatic disease. Of these, 34.37% exhibited bone and soft tissue metastases, while 65.63% had visceral metastases.

Regarding breast cancer subtypes, 56.25% had luminal A tumors, and 43.75% had luminal B tumors. All patients had at least one comorbidity, with 76.87% having one additional condition besides cancer, 19.37% having two or more comorbidities, and only 3.76% having no comorbidities. The most common comorbidities were diabetes, hypertension, cardiovascular conditions, and immunological disorders.

Performance status (PS) was evaluated using the ECOG scale, with 25% of patients classified as ECOG 0, 50.63% as ECOG 1, and 24.37% as ECOG 2. Additionally, all patients underwent evaluation using the G8 geriatric screening tool. A total of 54.37% of patients had a G8 score above 14, indicating no vulnerability, while 45.63% were classified as frail with G8 scores between 9 and 14, for whom comprehensive geriatric assessment (CGA) was recommended. CGAs were actually performed on 18 patients belonging to the group with a G8 score between 9 and 14, corresponding to 24% of this group.

Endocrine therapy combined with CDK4/6 inhibitors included aromatase inhibitors (letrozole or anastrozole) in 62.50% of patients and fulvestrant in 37.5%. Regarding the distribution of CDK4/6 inhibitors, 36.25% of patients received palbociclib, while abemaciclib and ribociclib were administered to 30.62% and 33.13% of patients, respectively. The survival outcomes demonstrated that, after a median follow-up of 25 months the median PFS for the cohort was 15 months, and the OS was 19 months. Figure 1 and Figure 2 present the PFS and OS data, respectively.

The multivariable Cox regression analysis for PFS, shown in Table 2, revealed significant prognostic factors for PFS in elderly patients with HR+/HER2- metastatic breast cancer treated with CDK4/6 inhibitors. Age at diagnosis did not significantly impact the PFS (HR: 1.06, 95% CI: 0.97–1.16, and *p* = 0.185). Similarly, BMI showed no significant effect on the PFS (HR: 0.99, 95% CI: 0.55–1.76, and *p* = 0.963). However, the luminal B subtype emerged as a significant prognostic factor, with a hazard ratio of 4.16 compared to luminal A (95% CI: 2.11–8.21, *p* < 0.001), indicating a worse prognosis for patients with the luminal B subtype. The presence of visceral disease did not significantly influence the progression-free survival (PFS) (HR: 1.04, 95% CI: 0.61–1.78, and *p* = 0.874) nor did the G8 score (HR: 0.79, 95% CI: 0.42–1.49, and *p* = 0.470). An ECOG performance status of 2 showed a trend towards a poorer PFS but was not statistically significant (HR: 1.89, 95% CI: 0.93–3.83, and *p* = 0.077).

Regarding treatment specifics, neither the type of hormone therapy (fulvestrant vs. AI; HR: 0.67, 95% CI: 0.32–1.36, and *p* = 0.267) nor the type of CDK4/6 inhibitor used (ribociclib vs. palbociclib, HR: 1.51, 95% CI: 0.74–3.07, and *p* = 0.255; abemaciclib vs. palbociclib, HR: 0.97, 95% CI: 0.50–1.85, and *p* = 0.918) showed significant differences in the PFS outcomes (Table 2).

Initially, 53.75% of patients received the standard dose of CDK4/6 inhibitors, while 46.25% started on a reduced dose. Dose reductions were necessary for 39.37% of patients due to grade 3 or higher toxicities.

The details of hematological and non-hematological toxicities are shown in Table 3. Among the 160 total patients treated with abemaciclib, ribociclib, and palbociclib, adverse events were reported with varying frequencies. Regarding anemia, 5 patients (3.2%) experienced grade ≥ 3 anemia, with the highest incidence observed in the palbociclib group, where 4 out of 58 patients (6.9%) were affected. In contrast, no cases of grade ≥ 3 anemia were reported in the ribociclib arm. Neutropenia was more common, with 68 patients (42.5%) experiencing the event, including 32 cases of grade ≥ 3 (20%). The incidence of neutropenia was particularly high among those treated with palbociclib, where 34 patients (58.7%) were affected, and 15 patients (25.9%) reported grade ≥ 3. Ribociclib also showed a significant rate of grade ≥ 3 neutropenia, affecting 10 patients (8.9%). Thrombocytopenia was relatively rare, with only three patients (1.9%) affected. Palbociclib presented the highest number of cases, with three patients (5.2%) reporting any-grade thrombocytopenia, but none of the patients treated with abemaciclib or ribociclib experienced this adverse event. Asthenia was reported by 38 patients (23.8%), with 13 patients (8.1%) experiencing grade ≥ 3. The occurrence of any-grade asthenia was highest in the ribociclib group, where 16 patients (30.2%) were affected. Grade ≥ 3 asthenia was more frequently seen in the abemaciclib group, with seven patients (14.3%) reporting this condition. Diarrhea was notably prevalent in patients treated with abemaciclib, affecting 25 patients (51.1%), with 7 patients (5%) reporting grade ≥ 3. In contrast, no cases of diarrhea were reported among the patients treated with ribociclib or palbociclib. ALT/AST elevation occurred in five patients (3.3%), with grade ≥ 3 events recorded in two patients (1.3%). The rates of ALT/AST elevation were consistent across the three treatments, with palbociclib showing three cases (5.2%) of any-grade events. Finally, QTc prolongation was a rare event, affecting only patients treated with ribociclib, with two patients (3.8%) experiencing grade ≥ 3 QTc prolongation.

## 4. Discussion

Although CDK4/6 inhibitors have been established in clinical practice based on positive outcomes from randomized clinical trials, their safety and effectiveness in real-world populations, particularly in elderly patients, are still being evaluated. In this multicenter, retrospective Italian analysis, CDK4/6 inhibitors combined with endocrine therapy were shown to be beneficial and safe as a first-line treatment in elderly patients with metastatic breast cancer in a real-world setting.

The subgroup analysis of elderly patients (≥65 years) from the first-line randomized international trials PALOMA-2, MONALEESA-2, and MONARCH-3 demonstrated that the combination of hormone therapy plus CDK4/6 inhibitors significantly improves both the PFS and the OS compared to hormone therapy alone [18].

In the PALOMA-2 trial, patients aged 65–74 years who received palbociclib plus letrozole showed a significant improvement in PFS compared to letrozole alone, with a median PFS of 27.5 months versus 21.8 months (HR 0.66; 95% CI 0.45–0.97; and *p* = 0.016). For patients aged ≥75 years, the median PFS for those on palbociclib was not reached, compared to 10.9 months for those on letrozole alone (HR 0.31; 95% CI 0.16–0.61; and *p* < 0.001).

In the MONALEESA-2 trial PFS was significantly improved in the ribociclib group for both older patients (HR 0.61; 95% CI 0.39–0.94) and those younger than 65 (HR 0.52; 95% CI 0.38–0.72). Overall survival (OS) also increased regardless of age.

The MONARCH-2 and MONARCH-3 trials showed that CDK4/6 inhibitors improve PFS in patients over 65 years, with no significant differences compared to younger patients. In MONARCH-2, the hazard ratios for PFS were similar across the age groups, with an interaction *p*-value of 0.695, and, in MONARCH-3, the interaction *p*-value was 0.634.

In our cohort, the results demonstrated that the median PFS was 15 months, while the median OS was 19 months. Overall, these data suggest that patients treated with CDK4/6 inhibitors in real-world clinical practice may have a worse prognosis than those selected for clinical trials, such as those in the pivotal studies which led to the approval of these drugs. These trials often include more rigorously defined selection criteria, excluding patients with severe comorbidities, poor performance status (ECOG PS >2), and impaired G8 scores.

Our survival outcomes, while lower than those reported in randomized clinical trials [5,6,7,8,9,10,11,12,13,14,15,16], are closely aligned with real-world studies such as the HeLLENIC Cooperative Oncology Group (HeCOG), suggesting that these results are representative of the broader, unselected patient population [19].

The multivariate analysis revealed significant findings regarding prognostic factors for PFS in elderly patients with HR+/HER2- MBC treated with CDK4/6 inhibitors. The luminal B subtype was significantly associated with a poorer prognosis, highlighting its importance in determining patient outcomes. Other factors such as age, BMI, visceral disease presence, G8 score, ECOG status, hormone therapy type, and CDK4/6 inhibitor type did not show significant associations with PFS in this cohort. These findings underscore the need for further research to understand the impact of these variables on treatment efficacy in elderly patients with metastatic breast cancer. Further research could also help identify which patients would absolutely benefit from CDK4/6 inhibitors, ensuring that this treatment is optimally targeted to those who need it most.

Regarding tolerability in this patient group, our analysis confirms the safety profile of CDK4/6 inhibitors in elderly patients. While AEs such as anemia, neutropenia, and asthenia were observed with varying frequencies, the majority were manageable, and only a small proportion of patients experienced grade ≥ 3 events. Importantly, the data suggest that, despite the occurrence of these AEs, severe or life-threatening events were relatively uncommon.

Anemia, while present in 8.7% of patients, remained relatively mild across treatments. Ribociclib, in particular, showed the lowest rates of severe anemia, which suggests that it may be better tolerated in terms of hematologic toxicity in certain patient populations. This is crucial in elderly patients who may already have underlying vulnerabilities such as pre-existing anemia or other comorbidities affecting bone marrow function.

Neutropenia, particularly grade ≥3, was the most frequent adverse event, affecting 20% of patients, with palbociclib showing the highest incidence. However, neutropenia is a well-recognized, manageable side effect of CDK4/6 inhibitors and is often treated effectively with dose adjustments. Elderly patients, who are often more susceptible to infections, must be monitored closely, but the manageability of this AE allows for continued therapy with proper precautions.

Thrombocytopenia was rare, and severe cases were almost non-existent, reinforcing the overall safety of these agents with respect to platelet suppression. This is especially beneficial for older patients who may be at higher risk of bleeding complications.

Asthenia, a common symptom in patients with cancer and often exacerbated by treatment, affected 23.8% of patients, with only 8.1% experiencing severe cases. This is a manageable side effect, especially in elderly populations, as dose modifications and supportive care can alleviate fatigue. Among the three drugs, ribociclib had the highest rates of asthenia, but these rates remained within an acceptable range for most patients.

Diarrhea was notably more prevalent in patients treated with abemaciclib (51.1%), a known side effect of this agent. However, despite its frequency, the incidence of severe grade ≥ 3 diarrhea was relatively low at 5%. The absence of diarrhea in patients treated with ribociclib and palbociclib further supports the idea that these agents may be preferable for older patients who are more prone to dehydration and electrolyte imbalances.

Liver enzyme elevation (ALT/AST) and QTc prolongation were rare across the cohort, with only 1.3% of patients experiencing significant changes. These low rates are encouraging, as elderly patients may have pre-existing cardiovascular and hepatic conditions that could be exacerbated by treatment. The minimal incidence of these severe AEs underscores the relative safety of these drugs even in more vulnerable populations.

The favorable safety profile observed in our cohort may be largely attributable to the fact that 46.25% of patients initiated CDK4/6 inhibitor therapy at a reduced dose. The correlations between frequent treatment interruptions in frail patients and overall clinical outcomes are currently being evaluated in a separate analysis, which will be reported in a future publication. This proactive dose adjustment likely contributed to the lower incidence of severe adverse events (grade ≥ 3) seen in this study compared to randomized clinical trials, where standard dosing is predominantly employed. Nearly 40% of patients required further dose reductions due to toxicity, which suggests that initiating treatment at a lower dose may have played a pivotal role in mitigating adverse effects without significantly compromising therapeutic efficacy. These findings underscore the critical importance of individualized dosing strategies in elderly patients with HR+/HER2- metastatic breast cancer, particularly in those with comorbidities or frailty.

Moreover, in this cohort of patients over 70 years old, no locoregional treatments, such as surgery or radiotherapy, were performed. While locoregional therapies may be considered in oligometastatic disease, there is currently a lack of definitive data on outcomes in this specific population, also because metastatic patients are less frequently managed by a multidisciplinary team compared to patients with early-stage disease [20].

Our study has certain limitations: First, the analysis was conducted retrospectively. Then, given the observational design, we cannot rule out the presence of both known and unknown confounders, nor can they be fully measured or controlled. The study was also conducted at a limited number of investigational sites in Italy, which may reduce the extent to which the findings can be applied to other medical centers or regions across Italy.

## 5. Conclusions

This retrospective analysis highlights the effectiveness and general safety of CDK4/6 inhibitors combined with endocrine therapy for elderly patients with HR+/HER2- metastatic breast cancer in real-world settings. Although the PFS and OS rates were lower than those in clinical trials, likely due to a more heterogeneous patient population, CDK4/6 inhibitors were effective across the various subgroups. The luminal B subtype was identified as a negative prognostic factor for PFS, while other factors, such as age and ECOG status, had no significant impact. Dose reductions due to toxicities were common, underscoring the need for individualized dosing and careful monitoring in frail patients. Further research is needed to refine treatment strategies for this population.

## Figures and Tables

**Figure 1 cancers-16-03442-f001:**
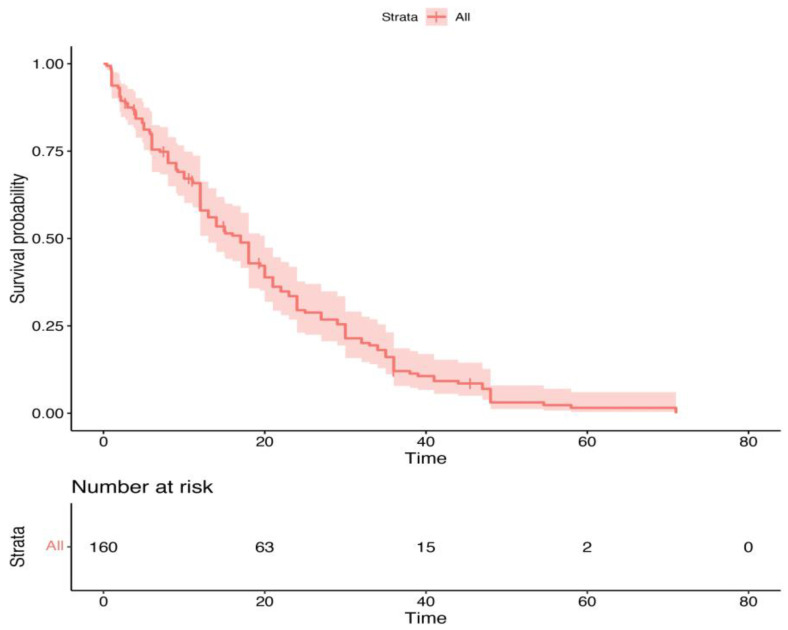
Kaplan–Meier progression-free survival curve for the entire cohort of patients receiving CDK4/6 inhibitor therapy.

**Figure 2 cancers-16-03442-f002:**
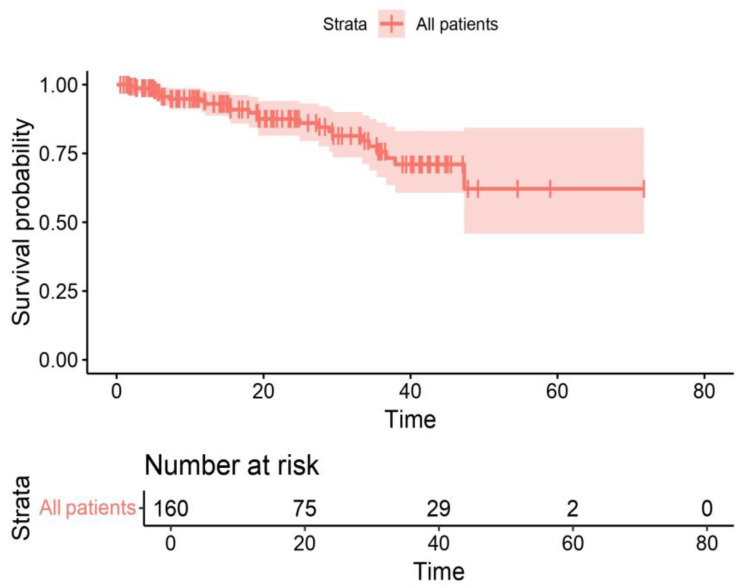
Kaplan–Meier overall survival (OS) curve for the entire cohort of patients receiving CDK4/6 inhibitor therapy.

**Table 1 cancers-16-03442-t001:** Clinical and demographic characteristics of patients undergoing CDK4/6 inhibitor therapy.

N° of Patients out of 160 (%)	Characteristics
	Age at the start of CDK4/6i therapy
82 (51.26%)	70–75
78 (48.74%)	>75
	Age at the diagnosis
89 (55.63%)	≤75
71 (44.37%)	>75
	Stage at the diagnosis
51 (31.87%)	I–III
109 (68.13%)	IV
	N° metastasis sites
55 (34.37%)	Soft tissue/bone
105 (65.63%)	Visceral
	BC subtype
90 (56.25%)	Luminal A
70 (43.75%)	Luminal B
	Comorbidities
6 (3.76%)	0
123 (76.87%)	1
31 (19.37%)	2 or more
	ECOG PS
40 (25.00%)	0
81 (50.63%)	1
39 (24.37%)	2
	G8 score
87 (54.37%)	>14
73 (45.63%)	≤14
	Endocrine therapy
100 (62.50%)	Letrozole or Anastrozole
60 (37.50%)	Fulvestrant
	CDK4/6i
49 (30.62%)	Abemaciclib
58 (36.25%)	Palbociclib
53 (33.13%)	Ribociclib
	Starting dose
86 (53.75%)	Standard
74 (46.25%)	Reduced
	Dose reduction
61 (38.13%)	Yes
99 (61.87%)	No
	Toxicities
35 (21.87%)	G1
44 (27.50%)	G2
46 (28.75%)	G3
75 (46.87%)	Temporary suspensions
28 (17.5%)	1
23(14.37%)	2
30 (18.75%)	3
	BMI
60 (37.5%)	BMI < 25
100 (62.5%)	BMI ≥ 25
15	MEDIAN PFS
19	MEDIAN OS

**Table 2 cancers-16-03442-t002:** Multivariate analysis of factors affecting progression-free survival (PFS).

*p*	95% CI	HR			Variables
0.185	0.97–1.16	1.06			Age *
0.963	0.55–1.76	0.99	≥25 vs. <25		BMI
<0.001	2.11–8.21	4.16	B vs. A		Luminal subtype
0.874	0.61–1.78	1.04	Yes vs. No		Visceral disease
0.470	0.42–1.49	0.79	≤14 vs. >14		G8 score
0.077	0.93–3.83	1.89	2 vs. 0–1		ECOG PS
0.267	0.32–1.36	0.67	Fulvestrant vs. AI		Type of hormone therapy
0.255	0.74–3.07	1.51	vs. Palbociclib	Ribociclib	Type of CDK4/6inh
0.918	0.50–1.85	0.97		Abemaciclib	

* Considered as continuous variable.

**Table 3 cancers-16-03442-t003:** Incidence and severity of adverse events across CDK4/6 inhibitors.

AEs	Any Grade No (%)			Grade ≥ 3 No (%)			Total Patients No 160 (%)	
	Abemaciclib No 49	Ribociclib No 53	Palbociclib No 58	Abemaciclib No 49	Ribociclib No 53	Palbociclib No 58	Any Grades	Grade ≥ 3
**Anemia**								
No	44 (89.7)	51 (96.2)	51 (87.9)	48 (97.9)	53 (100)	54 (93.1)	146 (91.3)	155 (96.8)
Yes	5 (10.3)	2 (3.8)	7 (12.1)	1 (2.1)	0 (0)	4 (6.9)	14 (8.7)	5 (3.2)
**Neutropenia**								
No	38 (77.5)	29 (54.7)	24 (41.3)	42 (85.7)	43 (81.1)	43 (74.1)	92 (57.5)	128 (80.0)
Yes	10 (22.5)	24 (45.3)	34 (58.7)	7 (14.3)	10 (8.9)	15 (25.9)	68 (42.5)	32 (20.0)
**Thrombocytopenia**								
No	49 (100)	53 (100)	55 (94.8)	49 (100)	53 (100)	56 (96)	157 (98.1)	158 (98.7)
Yes	0 (0)	0 (0)	3 (5.2)	0 (0)	0 (0)	2 (4)	3 (1.9)	2 (1.3)
**Asthenia**								
No	37 (75.5)	37 (69.8)	48 (82.7)	42 (85.7)	49 (92.4)	56 (96.5)	122 (76.2)	147 (91.9)
Yes	12 (24.5)	16 (30.2)	10 (17.3)	7 (14.3)	4 (7.6)	2 (3.5)	38 (23.8)	13 (8.1)
**Diarrhea**								
No	24 (48.9)	53 (100)	58 (100)	42 (85)	53 (100)	58 (100)	135 (84.4)	153 (95.7)
Yes	25 (51.1)	0 (0)	0 (0)	7 (5)	0 (0)	0 (0)	25 (15.6)	7 (4.3)
**ALT/AST increased**								
No	49 (100)	51 (96.2)	55 (94.8)	49 (100)	52 (98)	57 (98)	155 (96.7)	158 (98.7)
Yes	0 (0)	2 (3.8)	3 (5.2)	0 (0)	1 (2)	1 (2)	5 (3.3)	2 (1.3)
**Qtc prolongation**								
No	49 (100)	51 (96.2)	58 (100)	49 (100)	51 (96.2)	58 (100)	158 (98.7)	158 (98.7)
Yes	0 (0)	2 (3.8)	0 (0)	0 (0)	2 (3.8)	0 (0)	2 (1.3)	2 (1.3)

## Data Availability

Data is not available due to ethical or privacy restrictions.

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
