# Peer review of "Evaluating CDK4/6 Inhibitor Therapy in Elderly Patients with Metastatic Hormone Receptor-Positive, HER2-Negative Breast Cancer: A Retrospective Real-World Multicenter Study"

_cancers, 2024, doi:10.3390/cancers16203442_

Round 1

Reviewer 1 Report

Comments and Suggestions for Authors

The current manuscript indeed develops real-world insights in CDK4/6 inhibitors combined with endocrine therapy in elderly patients ≥70 years with HR+/HER2- metastatic breast cancer. It reports a median PFS of 15 months and OS of 19 months based on data from 160 patients in multiple centers in Italy.

These results are consistent with observations in real-life studies, although survival outcomes are moderately lower when compared with clinical trials.

The luminal B subtype was identified in this study as a very strong predictive factor for poorer PFS, while other factors such as age, BMI, visceral metastases, and type of hormone therapy did not show any significant influence on results.

The safety profile was consistent with known CDK4/6 inhibitor effects, though nearly 40% of patients required dose reductions due to toxicities.

Notwithstanding the strengths identified, there are some limitations that must be addressed.

Strengths:
Relevant Topic: The manuscript provides key real-world data in the use of CDK4/6 inhibitors in elderly patients with HR+/HER2- metastatic breast cancer.

Multicenter Study: Data is drawn from seven Italian centers, enhancing applicability in general clinical practice.

Prognostic Factor Identification: The luminal B subtype is recognized as a significant factor for poor PFS, hence providing helpful insight to clinicians in deciding appropriate treatment approaches.

Major flaws:

Retrospective Design: Being a retrospective study, it is prone to some potential confounders which cannot be fully controlled or measured. There is a need for a more in-depth discussion on how confounders were handled. This limitation has been already addressed by the authors in the limitations section. But, many more limitations are present! Moreover, was a power analysis performed? Please address this in Methods.

Treatment options: They are not discussed in details. Even in frail patients, surgery can be implemented in some very selected cases with promising oncological results. Cite PMID: 36551722 to improve your Discussion.

Frail Patients: The authors should further discuss how frail patients were managed, especially those who needed frequent treatment interruptions, and its implications on overall outcomes.

Longer follow-up may be needed to understand OS outcomes better, as in the Kaplan-Meier analysis, many patients were censored.

Minor flaws:

Please modify the title, it looks like it has been created with AI

Re-write your conclusions. They are too long and not very effective to readers.

Author Response

Dear Reviewer,

Thank you for your comments. Please find below the responses to the comments and the corresponding text changes (in yellow in the text) made accordingly.

Major flaws:

Comment1: Being a retrospective study, it is prone to some potential confounders which cannot be fully controlled or measured. There is a need for a more in-depth discussion on how confounders were handled. This limitation has been already addressed by the authors in the limitations section. But, many more limitations are present! Moreover, was a power analysis performed? Please address this in Methods.

Response 1:  We acknowledge that retrospective studies are inherently subject to confounders that cannot be fully controlled. As mentioned in the limitations section, we took several steps to mitigate the impact of confounders, such as adjusting for key clinical variables, though we recognize this as a limitation of our study.

Regarding your question on the power analysis, we confirm that a formal power analysis was not performed. The study was based on the available data from real-world clinical practice across multiple centers, which determined the sample size. We have now clarified this point in the Methods section.

Comment 2. Treatment options: They are not discussed in details. Even in frail patients, surgery can be implemented in some very selected cases with promising oncological results. Cite PMID: 36551722 to improve your Discussion.

Response 2 In the study no locoregional therapies, such as surgery or radiotherapy, were performed in any of the cases. Although oligometastatic disease (managed with surgery or radiotherapy) can be considered in some circumstances, there is a lack of definitive data on the outcomes in this specific population. Additionally, metastatic patients are rarely managed by a multidisciplinary team, unlike those with early-stage disease. We have added a brief clarification to the Discussion section.

Refrence 18 was added (Tinterri C, Sagona A, Barbieri E, Di Maria Grimaldi S, Jacobs F, Zambelli A, Trimboli RM, Bernardi D, Vinci V, Gentile D. Loco-Regional Treatment of the Primary Tumor in De Novo Metastatic Breast Cancer Patients Undergoing Front-Line Chemotherapy. Cancers (Basel). 2022 Dec 17;14(24):6237. doi: 10.3390/cancers14246237. PMID: 36551722; PMCID: PMC9777012)

Comment 3. Frail Patients: The authors should further discuss how frail patients were managed, especially those who needed frequent treatment interruptions, and its implications on overall outcomes.

Response 3: These correlations are currently being explored in a separate analysis specifically aimed at evaluating the impact of treatment interruptions on survival and other clinical outcomes. The findings from this analysis will be reported in a future publication. The text has been changed

Minor flaws:

Comment 4: Please modify the title, it looks like it has been created with AI

Response 4: the title has been modified as follows: Evaluating CDK4/6 Inhibitor Therapy in Elderly Patients with Metastatic Hormone Receptor-Positive, HER2-Negative Breast Cancer: A Retrospective Multicenter Study

Comment 5:Re-write your conclusions. They are too long and not very effective to readers.

Response 5: done.

Reviewer 2 Report

Comments and Suggestions for Authors

Major points:

·        Discussion about the evolving landscape within HER2 negative patient population should be added in the paper. Authors are encouraged to include HER2 IHC data, if possible. In line with that, a deep discussion about HER2 low expression should be prepared including the clinical impact on prognosis and on the emerging therapies in this space.

·        Other combinations with CDK4/6 should also be addressed (e.g., tamoxifen, oral SERDs, the addition of PIK3CA/AKT/mTOR inhibitors in the combo, etc) and implications in the elderly patient population. Why data on patients receiving exemestane were not included in the paper? Comments about those points will enrich the discussion.

·        Regarding the recurrent patients (31.87%), what was the proportion of prior CDK4/6i? Comments on sequencing would also increase relevance of the paper.

Minor points.

·        As the concept of HR-low is also evolving rapidly in terms of prognostic value, that may be another area to discuss, though this is more related to younger population. Are there IHC results on this?

·        I know the term “real-life” can be used and there is no consensus on this. However, as a minor comment, I would favor the term “real-world” in the title, especially for paper visibility/searchability purposes.

Comments on the Quality of English Language

n/a

Author Response

Dear Reviewer,

Thank you for your comments. Please find below the responses to the comments and the corresponding text changes (in orange in the text) made accordingly.

Major flaws:

Comment1:  Discussion about the evolving landscape within HER2 negative patient population should be added in the paper. Authors are encouraged to include HER2 IHC data, if possible. In line with that, a deep discussion about HER2 low expression should be prepared including the clinical impact on prognosis and on the emerging therapies in this space.

Response 1:  We have incorporated the discussion on the evolving landscape within the HER2-negative patient population into the introduction, as requested. We have specified in the Materials and Methods: Study Design and Patients section that all patients in the study presented with HER2-negative disease. However, since this analysis was conducted prior to the identification of HER2-low as a distinct category, HER2 status was recorded as either positive or negative in patient records. Therefore, information regarding HER2 2+, 1+, or 0 expression levels is not available for this cohort.

Comment 2. Other combinations with CDK4/6 should also be addressed (e.g., tamoxifen, oral SERDs, the addition of PIK3CA/AKT/mTOR inhibitors in the combo, etc) and implications in the elderly patient population. Why data on patients receiving exemestane were not included in the paper? Comments about those points will enrich the discussion.

Response 2 We have addressed the point regarding other combinations with CDK4/6 inhibitors, such as tamoxifen, oral SERDs, and PIK3CA/AKT/mTOR inhibitors, as well as the implications for elderly patients, in the introduction section.

Additionally, we do not have data on patients receiving exemestane because none of the patients in our cohort received the combination of exemestane and CDK4/6 inhibitors.

Comment 3. Regarding the recurrent patients (31.87%), what was the proportion of prior CDK4/6i? Comments on sequencing would also increase relevance of the paper.

Response 3: None of the patients had received CDK4/6 inhibitors in the adjuvant setting. This information has been specified in the section 2. Materials and Methods 2.1 Study Design and Patients

Minor flaws:

Comment 4: As the concept of HR-low is also evolving rapidly in terms of prognostic value, that may be another area to discuss, though this is more related to younger population. Are there IHC results on this?

Response 4: we appreciate the suggestion to discuss this area further, however, in our current study, we do not have IHC results specific to HR-low expression. This is an important topic that warrants further investigation in future research to better understand its implications for prognosis and treatment outcomes

Comment 5:I know the term “real-life” can be used and there is no consensus on this. However, as a minor comment, I would favor the term “real-world” in the title, especially for paper visibility/searchability purposes.

Response 5: In response to your feedback, we have changed the title to use the term 'real-world' for improved visibility and searchability 

Round 2

Reviewer 1 Report

Comments and Suggestions for Authors

The manuscript can be accepted in the present form.